# Preterm Infants’ Airway Microbiome: A Scoping Review of the Current Evidence

**DOI:** 10.3390/nu16040465

**Published:** 2024-02-06

**Authors:** Sofia Fatima Giuseppina Colombo, Chiara Nava, Francesca Castoldi, Valentina Fabiano, Fabio Meneghin, Gianluca Lista, Francesco Cavigioli

**Affiliations:** 1Department of Pediatrics, Buzzi Children’s Hospital, 20154 Milan, Italy; sofia.colombo@unimi.it (S.F.G.C.); chiara.nava@unimi.it (C.N.); 2Division of Neonatology, Buzzi Children’s Hospital, 20154 Milan, Italy; francesca.castoldi@asst-fbf-sacco.it (F.C.); fabio.meneghin@asst-fbf-sacco.it (F.M.); gianluca.lista@asst-fbf-sacco.it (G.L.);

**Keywords:** preterm infants, lung microbiome, airway microbiome

## Abstract

The aim of this scoping review was to investigate and synthesize existing evidence on the airway microbiome of preterm infants to outline the prognostic and therapeutic significance of these microbiomes within the preterm population and identify gaps in current knowledge, proposing avenues for future research. We performed a scoping review of the literature following the Arskey and O’Malley framework. In accordance with our inclusion criteria and the intended purpose of this scoping review, we identified a total of 21 articles. The investigation of the airway microbiome in preterm infants has revealed new insights into its unique characteristics, highlighting distinct dynamics when compared to term infants. Perinatal factors, such as the mode of delivery, chorioamnionitis, the respiratory support, and antibiotic treatment, could impact the composition of the airway microbiome. The ‘gut–lung axis’, examining the link between the lung and gut microbiome as well as modifications in respiratory microbiome across different sites and over time, has also been explored. Furthermore, correlations between the airway microbiome and adverse outcomes, such as bronchopulmonary dysplasia (BPD), have been established. Additional research in neonatal care is essential to understand the early colonization of infants’ airways and explore methods for its optimization. The critical opportunity to shape long-term health through microbiome-mediated effects likely lies within the neonatal period.

## 1. Introduction

The airway microbiome is defined as the collective genome of the microbial community that colonizes the respiratory tract [1]. Until recently, the infant airway was thought to be sterile. It is now known that the respiratory tract is colonized by microbiota, with reports of as many as 10–100 bacterial cells per 1000 human cells, but at present, the normal composition of the neonatal respiratory microbiome is not fully understood [2]. The lung microbiome observed at birth undergoes changes during the first week of life, leading to the establishment of one or more dominant organisms by day of life (DOL) 7, which continue to evolve over the subsequent months of postnatal life [3]. There is limited understanding of the mechanisms governing the development and progression of the airway microbiome in premature infants. The environment around the fetus, including the amniotic fluid, placenta, and vagina, has a distinct microbiome that could potentially be introduced into the newborn’s respiratory tract before or during birth. While it was previously believed that infants delivered vaginally acquire maternal vaginal flora and those born via cesarean section are primarily colonized with maternal skin flora, it now seems that all fetuses experience initial colonization during the prenatal period [3,4], although the ‘sterile womb’ hypothesis has not yet been completely disproven [5]. There are several factors that can influence the composition and development of the respiratory microbiome, especially in preterm infants. These factors include the mode of delivery, the environmental microbiome surrounding the fetus, early exposure to prenatal and postnatal antibiotics and steroids, sepsis, the use of mechanical ventilation or other respiratory support devices, feeding and nutrition, and the concomitant intestinal microbiota development [6]. Indeed, both the gut and the respiratory tracts share a common embryonic origin, and there appears to be a connection between the development of the gut and lung microbiomes (gut–lung axis) [7,8]. The composition of the oral and lung microbiome may also be affected by the administration of oropharyngeal colostrum to the buccal mucosa of preterm infants [9,10,11].

The maturation of early-life microbiota during the perinatal period plays a key role in shaping the immunological, neural, and physiological development of infants, ultimately affecting their overall health [6]. Lung and airway microbial colonization normally occurs in the prenatal or postnatal period and can be involved in normal immune and structural development. The interplay between commensal bacteria and lung epithelial cells, along with leukocytes, triggers the activation of toll-like receptors, leading to the stimulation of dendritic cells and the initiation of TH1- and TH17-mediated responses. This promotes normal immune tolerance. In contrast, disrupting or eliminating this normal commensal microbiota can potentially trigger atypical inflammatory reactions (such as mixed Th1/Th2/Th17–type response with high levels of IL-1β, TNF-α, macrophage inflammatory protein-1β, which may lead to chronic inflammation), which could contribute to the development of bronchopulmonary dysplasia (BPD) [3,6,12]. Airway dysbiosis could lead not only to the development of BPD but also to a higher incidence of asthma and pneumonia in childhood [13].

Considering all that has been said so far, the objectives of this review are to (1) research and summarize the available evidence regarding the airway microbiome of preterm infants, (2) describe the prognostic and therapeutic role of these microbiomes in the preterm population, (3) identify knowledge gaps in this topic and thus suggest considerations for future research. As such, we conducted a scoping review of the relevant articles in the subject of lung microbiome in preterm infants.

## 2. Materials and Methods

The study protocol of this review was developed following the Preferred Reporting Items for Systematic Reviews and Meta-Analyses Extension for Scoping Review (PRISMA-ScR) guidelines [14] and following the methodological guidance reported in Arksey and O’Malley’s and Levac’s papers [15,16].

The systematic literature search was performed in July 2023 (and updated in November 2023) on the following databases: Web of Science, Cinahl, Ovid Medline, and Embase. Search strategies and keywords for each database are reported in Appendix A. Additional studies were identified by screening the reference lists of the full-text screened articles.

Given that scoping reviews are made in order to “map”, summarize, and clarify emerging evidence about a topic, they typically have broader inclusion criteria. Thus, we included in our review any type of human study (randomised controlled trial (RCT), prospective and retrospective cohort studies, case reports) reporting on airway microbiota of infants born prematurely (before 37 weeks’ gestation), with first samples during the neonatal period. We only included those studies using culture-independent molecular techniques and reporting at least one alpha- or beta-diversity metric or a microbial profile. We did not focus on any particular outcome in order to gain a broader view about the role of airway microbiota on the pathophysiology of prematurity, on preterm outcomes, and on possible therapeutic interventions. We included studies regardless of publication date. We excluded animal studies, ongoing studies, non-English written studies, narratives, and systematic reviews.

Two authors independently screened all titles and abstracts obtained from the searches and assessed the full texts of all the potentially eligible studies. Conflicts were solved by discussion or by consultation with a third author. Afterward, two independent reviewers extracted study characteristics using a predefined charting form that was further refined. The relevant information extracted were author, year, country of publication, article type, study design, participant characteristics, type of microbiome, intervention, and outcomes. One author audited quality, the sources of evidence, and data charting to confirm the validity of the results.

## 3. Results

A total of 1123 references were retrieved through the search stage; in addition, 1 more eligible article was found by screening the reference lists, giving a final record count of 769 after duplicate removal. After title, abstract, and full-text screening, we included 21 studies, the characteristics of which are reported in Table 1. Please see Figure 1 for the flow diagram based on PRISMA.

The studies were published between April 2010 and November 2023. Eight studies were conducted in the US, four in China, four in the UK, three in Germany, one in Italy, one in Australia, and one in Switzerland and New Zealand.

### 3.1. Airway Microbiome Differences between Term and Preterm Infants

Some of the included studies described differences between the preterm and term infants’ airway microbiota. Pattaroni and co-authors [23] demonstrated that the gestational age has a strong effect on airway microbiota composition. In fact, studying the tracheal aspirates of 26 preterm and 19 term-born intubated infants, they found that the samples could be divided into three microbiota clusters: one dominated by *Staphylococcus*, one by *Ureaplasma* genera, and one characterized by a balanced composition, including *Streptococcus, Neisseria*, *Prevotella*, *Porphyromonas*, *Veillonella,* and *Fusobacterium* genera (the latter called “mixed”). Thus, they found that none of the samples collected from infants born before 30 weeks’ gestation (22 babies) clustered into the mixed profile. Otherwise, the samples of term infants mostly fell into the ‘mixed’ cluster. In 2022, McDavid and colleagues [21] confirmed postmenstrual age (PMA) as a central factor influencing mucosal microbiota. They conducted a longitudinal study evaluating the nasal and gut microbiome of 122 preterm and 80 full-term infants from birth until 12 months of age, evaluating a total of 1748 nasal swabs, and they found that lower gestational ages (GA) were associated with community state types (CST) dominated by *Staphylococcus* that later diverged to more niche-specific taxa, such as *Streptococcus* and *Corynebacterium*, or gastrointestinal taxa, such as *Enterobacteriales* and *Clostridiales.*

Similarly, Grier et al. [19] showed that infants born prematurely and term born are initially colonized by throat and nose microbiota that exhibit a distinct profile, which then tend to converge towards a similar pattern around 50 weeks PMA. In fact, the most represented CSTs in the earliest PMA were characterized by high levels of *Staphylococcus*, which decreased after 39 weeks PMA. After week 40 PMA, nose and throat microbiome were dominated by *Streptococcus*, with an emergence of *Corynebacterium*, *Alloiococcus*, *Moraxella*, and *Veillonella* in the nasal sites and *Veillonella* and *Neisseria* in the throat. More specifically, they showed that CSTs were influenced in different ways by prematurity: some community states were delayed (convergent), and some were over or under-represented depending on gestational age at birth (idiosyncratic to maturity at birth). Instead, some CSTs were entirely unaffected by prematurity, with occurrence depending entirely on postnatal age (chronological).

### 3.2. Airway Microbiome Temporal and Anatomical Modification

Another aspect that the included studies focused on was to describe the longitudinal changes of the respiratory microbiota of the preterm infants in the different parts of the respiratory tract (nose, throat, trachea, lungs) and in relation to other anatomical sites (mouth, gut, skin).

Grier et al. [19], in 2018, described and compared the patterns of the development of the nose, throat, and gut microbiota from birth through the first year of life of 82 preterm and full-term infants. They collected weekly samples from the preterm infants during hospital stays, monthly samples after their discharge, and samples from full-term neonates; in the case of acute respiratory illness, the sample was excluded. They clustered the samples into CSTs, which indicate approximate microbiota composition. Thus, they found that the CSTs of the three sampled sites are very different: the variability of taxa is low at the pharyngeal level (with a predominance of *Streptococcus* or *Staphylococcus*), increases at the nasal level (*Streptococcus* and *Corynebacterium* ranging from 5 to 50%) and is highest at the intestinal level (average abundance of each taxa of 1%).

These patterns change during the first year of life in a similar order in all infants, evolving from simpler to more diverse patterns within each body site. In particular, they showed how, in the earliest stages after birth, both gastrointestinal and respiratory microbiota contain high levels of *Staphylococcus*, which decrease beyond 39 weeks’ PMA. Thereafter, a transition to anaerobic germs occurs at the intestinal level, while at the respiratory level, an aerobic microbiota remains, dominated by *Streptococcus* (both in the nose and throat), with an emergence of *Corynebacterium*, *Alloiococcus*, *Moraxella*, and *Veionella* in the nose and *Veionella* and *Neisseria* in the throat.

A strong association between microbiota CSTs across body sites, in particular, between CSTs of the nose and gut and taxa in the throat, was shown by Grier et al. [19]. As the association cannot be entirely explained by time and by direct transmission between different sites, they hypothesized a possible systemic coordination during early-life development.

Similar observations were made by Gallacher and colleagues [18] regarding the temporal evolution of the microbiota. In 2020, they analyzed in a large prospective study a total of 1102 samples (nasopharyngeal aspirates (NPA), tracheal aspirate fluids (TAF), bronchoalveolar lavage (BAL), and stool) from 55 mechanically ventilated preterm infants (median gestational age (GA) 26.0 weeks). They found that the bacterial load was lower soon after birth (day 1–3) in the upper airways (with only 1.1% and 6.7% of TAF and NPA, respectively, successfully sequenced) than in the bowel (with 35.7% of success rate in stool samples). The bacterial load reached a peak around day 4–7 in BAL and TAF, slightly later (8–14 days) in NPA and stool. They showed Proteobacteria and Firmicutes being the most abundant phyla in all samples, with Tenericutes (*Ureaplasma* and *Mycoplasma*) also present in the respiratory samples. During the first month of life, the TAF and BAL microbiota pattern remained stable (with a relative abundance of Proteobacteria); instead, Proteobacteria abundance in NPA and stool increased from 15–30% to 46–55%, respectively. *Staphylococcus* was the dominant genus in nearly 30% of all samples. Nevertheless, they did not find any significant pattern over time, and all anatomical sites sampled showed their own distinct pattern of bacterial community.

Another small prospective study conducted in 2021 [3] evaluated 80 samples of oral and tracheal aspirates from 42 intubated preterm infants. Oral samples were collected at DOL 3 ± 1 and tracheal samples at DOL 3 ± 1 and 7 ± 1. They found *Staphylococus*, *Halomonas*, *Ureaplasma*, *Streptococcus*, *Bacillus*, *Nesterekonia*, *Schlegelella*, *Escherichia*, *Thermobacillus*, and *Acinetobacter* as the predominant general in overall oral and tracheal aspirates. Among the samples, *Shlegelella* has been found to be statistically more abundant in the early tracheal aspirates than in early oral and late tracheal samples. No other differences were found in the relative abundance of the other genera. Lastly, alpha diversity in the trachea at DOL 3 was statistically higher than that of saliva at DOL 3 and successively decreased to DOL 7, with the most prevalent genera approximating the early oral composition by DOL 7.

Some studies analyzed the microbial characteristics of the intestine and lung for exploring the mechanism of the gut–lung axis. Yang et al. [31] collected and analyzed stool and pharyngeal secretion samples on DOL 1 and on DOL 28 from 13 appropriate-for-gestational-age (AGA) preterm infants. They observed that, at DOL 1, there was no statistically significant difference in microbial composition between the intestine and pharynx, in contrast to what was observed on DOL 28. The pharyngeal microbiota was primarily composed of *Ureaplasma*, *Bacteroides*, and *Fusobacterium* on DOL 1 and *Streptococcus* and *Rothia* on DOL 28. In contrast, the gut microbiota predominantly consisted of *Unidentified Enterobacteriaceae*, *Ralstonia*, *Streptococcus*, *Fusobacterium*, and *Ureaplasma* on DOL 1 and of *Unidentified Clostridiales*, *Klebsiella*, *Unidentified Enterobacteriaceae*, *Enterobacter*, and *Streptococcus* on DOL 28. The microorganisms identified in both the intestine and pharynx were *Ureaplasma* and *Fusobacterium* on DOL 1 and *Streptococcus* on DOL 28. Similarly, Young et al. [33] reported that microbiomes of different sites (oral and endotracheal secretions, breast milk, and stool) in extremely preterm infants were most similar immediately after birth and diverged longitudinally over the first 60 days of life. Throughout the entire study period and across all sites, four dominant taxa were consistently noted: *Enterococcus*, *Enterobacteriaceae*, *Staphylococcus*, and *Escherichia*.

In 2023, Yao et al. [32] conducted a study on 10 twin pairs (20 cases) and 20 singleton preterm infants, collecting endotracheal suction after birth and stool samples from the first fetal stool. They found that the infants did not exhibit significant differences in gut or respiratory bacterial colonies, suggesting that the diversity and richness of the initially colonized bacteria were similar between twins and singletons. Bacterial colonies were more abundant and homogeneous in the gut than in the airways, as shown by an alpha-diversity analysis; the dominant intestinal phylum and genera were *Proteobacteria* and *Acinetobacter* and *Comamonas*, respectively. In contrast, the most abundant phylum and genera in the airways were Firmicutes and *Streptococcus* and *Ralstonia*, respectively.

### 3.3. Airway Microbiome and the Immune System

Among the studies we included in this scoping review, four evaluated the possible association between airway microbiome and local inflammatory response; the other two instead investigated the connections with immunological system development and the risk of sepsis.

About the inflammatory status of the airway, Gallacher et al. [18] in 2020 measured IL-6 and IL-8 in all samples and found that they were undetectable in NPA and stool; instead, they were present in high concentration in the lower airway samples (BAL and TAF) in which the sequencing was successful (sufficient bacterial load). Afterward, they evaluated temporal changes in the latter interleukin levels and showed that each patient has episodic increases of IL-6 and IL-8 corresponding to an increased bacterial load in the samples within 24 h and to the presence of a predominant taxonomic unit, such as *Acinetobacter*, *Enterobacteriaceae*, *Mycoplasma*, and *Ureaplasma*.

One year later, Brewer et al. [3] quantified leukocyte activation by polymerase chain reaction (PCR) in tracheal aspirates collected on DOL 3 of preterm infants, quantifying the expression of inflammatory genes (CD66, IL1b, ICAM1). They found a higher expression of IL1b in the samples with lower bacterial diversity. Instead, the levels of the expression of the latter genes were not associated with the need for longer intubation. Furthermore, Lohman et al. [1] reported that there were significant differences in the levels of IL-1β, IL-10 and tumor necrosis factor-α on endotracheal aspirate samples of premature infants exposed or not to chorioamnionitis. The levels of lipoteichoic acid (LTA) and lipopolysaccharide (LPS) were not statistically higher in infants exposed to chorioamnionitis, and the levels of all other biomarkers also did not differ significantly.

An interesting study by Wagner et al. [35] assessed the airway response in mechanically ventilated preterm infants by examining 12 proteins in tracheal aspirates collected at 7 days of age. Infants were categorized into three clusters based on protein profiles: Cluster 1 had high levels of most proteins, Cluster 2 showed a mix of high proinflammatory and low anti-inflammatory proteins, and Cluster 3 had low levels of all proteins. Associations were found between these biomarker clusters and antenatal factors. Additionally, subtle differences in airway microbiota were observed among the clusters, such that Cluster 3 infants (more exposed to pre-eclampsia) were more likely to be dominated by *Staphylococcus*, with a relative absence of *Ureaplasma*, and Cluster 1 (more exposed to antepartum haemorrhages) exhibited the highest levels of *Streptococcus* and *Gemella*. This could highlight how *Staphylococcus species* act as commensal organisms that do not elicit a significant inflammatory response, in contrast to *Ureaplasma*, which seems to be implicated in airway inflammation, as suggested by various studies [36,37,38,39].

Regarding sepsis, in a prospective multicenter cohort study, Dos Anjos Borges et al. [17], who analyzed the distribution of taxa in pharyngeal swabs between early-onset sepsis (EOS) and non-EOS neonates, observed that taxa with a higher relative abundance in non-EOS cases included common skin inhabitants, such as *Staphylococcus epidermidis* or *Cutibacterium acnes*.

Lastly, a recent study [21] demonstrated that preterm and term infants’ airway microbiota and immune system development are strictly correlated, constituting an age-independent microbiota–T cell axis. In particular, McDavid and co-authors found that early T cell phenotypic differentiation predicts later microbiota, and T cell functional maturation follows microbiota exposures. Thus, they explored abnormal developmental paths in newborn T cells and microbiota, revealing their association with respiratory issues in infancy. The study suggests that prenatal exposure to antibiotics or infections disrupts these trajectories, impacting respiratory microbial colonization and predicting the risk of respiratory compromise within the first year of life.

### 3.4. Airway Microbiome and Perinatal Factors

A number of potential clinical determinants of the airway microbiome (such as the mode of delivery, antibiotic exposure, sex, chorioamnionitis) were analyzed by many of the included studies. Gallacher et al. [18] found that preterm infants delivered vaginally had a significantly higher abundance of Gram-negative genera (*Acinetobacter*) and *Pseudomonas* and *Mycoplasma* in TAF samples and *Serratia* in NPA. Instead, a cesarean section showed increased *Staphylococcus* in NPA and TAF samples. An *Acinetobacter* presence was significantly higher in infants recruited by one of the two recruiting sites, which was having an outbreak of this bacteria, showing the great influence of the environment. No differences instead were noted between sex.

Similarly, Pattaroni et al. [23] found that *Ureaplasma* genera was only seen in tracheal aspirates of vaginally born preterm neonates and in a single case of emergency cesarean section with a membrane rupture; instead, *Staphylococcus* was predominant in the samples of babies born via a cesarean section. Moreover, they demonstrated that in the preterm population, the delivery mode has the highest impact (28%) on the variability of the microbiota composition. Differently, in the term-born group, this factor loses its effect, and the postnatal age explains the 14% of microbiota variance. Rosenboom et al. [25] reported that the significant impact of the birth mode on the airway microbiome at one week postpartum was no longer detectable one month postpartum.

In contrast, Lal et al. [20] highlighted that there were no statistically significant differences in the respiratory microbiome based on the mode of delivery (cesarean or vaginal delivery). Payne and colleagues [24] did not find a correlation between the mode of delivery and the rate of isolation of *Mycoplasma* and *Ureaplasma* spp., but denaturing gradient gel electrophoresis (DGGE) revealed that microbes were more commonly found in nasogastric aspirate (NGA) samples from preterm infants delivered vaginally compared to those delivered via cesarean section.

Tirone et al. [28] demonstrated that, while the vaginal microbiota of mothers with a spontaneous preterm birth (SP^PTB^) showed a significant difference in alpha diversity, marked by a decrease in *Lactobacillus* and an increase in *Proteobacteria* abundance, there were no significant differences in alpha e beta diversity between the neonatal bronchoalveolar lavage fluid (BALF) samples of the SP^PTB^ group and the medically indicated preterm birth (MI^PTB^) group. Regarding chorioamnionitis, Lohman et al. [1] also reported that there were no statistically significant differences in bacterial diversity in endotracheal aspiration samples from newborns with and without exposure to chorioamnionitis, although they noted a trend toward decreased diversity in those infants who had been exposed. In contrast, Lal et al. [20] noted a reduction in *Lactobacillus* spp. expression in infants who were exposed to chorioamnionitis.

Grier and co-authors [19] reported that samples taken during antibiotic therapy had a global suppression of bacterial growth in TAF and NPA, but no differences in alpha diversity with respect to those taken when not on antibiotics. Lal et al. [20] observed that the airway microbiome did not differ between infants born to mothers who received prenatal antibiotics and those born to mothers who did not. In addition, sixty percent of the Lohmann et al. [1] cohort received 48 h of empiric iv ampicillin and gentamicin, and there were no significant differences in Shannon’s index in the initial samples between infants receiving empiric antibiotics and those not, similarly as said by Pattaroni et al. [23]. In addition, at 15 months of age, the influence of antimicrobial therapy was not detectable [25].

In one study [3], the administration of oropharyngeal colostrum (OPC) to intubated premature infants was not correlated to any differences on bacterial composition nor on the inflammatory activity of oral and tracheal samples. Nevertheless, a delayed first administration was associated with a trend toward decreased alpha diversity and increased IL1b.

### 3.5. Airway Microbiome Functionality

Using the predictive functional profiling of microbial communities (PICRUSt), which analyzes the functional metagenome based on marker gene data, researchers [19] tried to predict the functions of microbiota in the rectum, nose, and throat over a baby’s first year. Aware of the limitations of PICRUSt [40], they analyzed the top pathways in different community state types (CSTs) at each site. Early CSTs in all three sites showed a distinct function enriched in pathways such as lipid, purine, and pyrimidine metabolism, crucial for neonatal growth and colonization. Energy metabolism in early and late CSTs was linked to the pentose phosphate pathway and later diversified with exposure to different diets. Two-component signal transduction pathways suggested increased communication within the microbiota community and with the host. Comparing functions in initial and later CSTs in the nose and gut revealed emerging pathways, indicating the development and diversification of microbial communities in response to environmental factors and changes in energy sources.

### 3.6. Airway Microbiome and BPD

In our literature review, about half of the included studies examined the role of the respiratory microbiome in the development of BPD.

In 2010, Payne et al. [24] described the airway microbiome of fifty-five preterm infants requiring mechanical ventilation by analyzing endotracheal and nasogastric aspirate (ETA, NGA) samples using a culture-independent approach, denaturing gradient gel electrophoresis (DGGE) profiling. In addition, they used species-specific PCRs for *Mycoplasma* and *Ureaplasma* spp. and correlated microbial findings and perinatal risk factors to the development of BPD. They discovered that bacterial sequences associated with coagulase-negative *Staphylococcus species* (*S. haemolyticus* and *S. epidermidis*) were the most prevalent. *Fusobacterium nucleatum* was the most frequently identified organism in NGA samples, while in ETA samples, it was *Staphylococcus haemolyticus*. Additionally, *Mycoplasma hominis* was detected through a PCR analysis in 9–13% of ETA samples and *Ureaplasma* spp. in approximately 50%. They noted a significant correlation between the identification of *Ureaplasma* spp. on ETA samples and a poor outcome for BPD. The presence of this microorganism was also correlated with a more prolonged period of mechanical ventilation.

A small observational study performed by Mourani and colleagues [22] in 2011 investigated the bacterial composition of the respiratory tract using pyrosequencing in 10 preterm infants (gestational age < 28 weeks, birth weight 500–1250 g) who had been mechanically ventilated for at least 21 days and later developed BPD. They collected tracheal aspirate samples from each patient at four time points (within 72 h of birth and at 7, 14, 21 days of life). Mourani et al. found that, in the first 72 h after birth, the majority of tracheal aspirates (8/10 samples) had an undetectable bacterial load. Within the first week of life, all samples collected contained a detectable bacterial load (>70 copies/reaction), and all were successfully amplified for bacterial identification. Almost all samples (96.9%) with bacterial detection had a dominant organism, most commonly either *Staphylococcus* spp. and *Ureaplasma* spp. *Pseudomonas aeruginosa*, *Enterococcus faecalis*, and *Escherichia coli* were the other dominant organisms identified in tracheal aspirate samples. Bacterial load, Shannon’s alpha-diversity index, and evenness were not associated with BPD severity or infant age. Stressman et al. [27], in a study regarding fourteen endotracheal secretion samples from eight mechanically ventilated preterm infants at risk of developing BPD, reported *Staphylococcus aureus*, *Enterobacter* spp., *Moraxella catarrhalis*, *Pseudomonas aeruginosa*, *Streptococcus* spp., and *Ureaplasma* spp. as the most common bacterial sequences. Due to the limited sample size (3/8 infants with BPD), they were unable to make a comparison of the airway microbiota between infants with and without BPD.

Wagner et al. [34], in 2017, also examined the relationship between the airway microbiota and the severity of BPD in mechanically ventilated preterm infants (gestational age ≤ 34 weeks, birth weight 500–1250 g) using 16S rRNA sequencing. A microbiome analysis was performed on tracheal aspirate samples of 152 infants (51, 49, and 52 with mild, moderate, and severe BPD, respectively) collected at enrollment (intubation) and 7, 14, and 21 days of age. The cross-sectional analysis conducted on the 7-days-of-life samples did not reveal a significant correlation between the microbiome composition and the subsequent severity of BPD. *Staphylococcus* (68%) and *Ureaplasma* (18%) spp. were predominant in the microbiome, but the Shannon diversity, bacterial load, and relative abundance of individual taxa showed no strong associations with BPD status. In contrast, a longitudinal analysis of airway microbiome composition over time demonstrated that preterm infants who eventually developed severe BPD experienced a greater turnover in bacterial communities with age acquired less *Staphylococcus* in the initial days after birth and had higher initial levels of *Ureaplasma* spp.

On the other hand, Xu et al. [29] demonstrated that the Shannon index of tracheal aspirate was significantly lower at birth (day 1) and on day 7 after birth in premature infants with BPD compared to those without BPD. The disparity was more pronounced on day 1 and exhibited a negative correlation with the severity of BPD. Additionally, they observed a significant difference in the bacterial composition of the airway microbiome at birth among the four groups (mild BPD, moderate BPD, severe BPD, non-BPD), with a less distinct difference on day 7 after birth. At the phylum level, Proteobacteria dominated the airway microbiome of all infants on day 1 and 7, with no significant differences in composition among the four groups. However, at the genus level, the composition of the airway microbiome differed significantly among the four groups on day 1. *Stenotrophomonas* was more abundant in BPD compared to non-BPD, and its abundance positively correlated with the severity of the disease.

Lohman et al. [1] were the first to analyze the airway microbiome of premature infants and investigate its correlation with bacterially mediated inflammation and the onset of BPD through 16S rRNA sequencing. They enrolled twenty-five patients with gestational age ranging from 24 to 32 weeks who underwent endotracheal intubation and mechanical ventilation in the first 24 h of life; ten of them developed BPD. Longitudinal tracheal aspirate samples were collected at the time of intubation (within 24 h of delivery) and at days of life 3, 7, and 28 if the infants were still intubated at those time points. Lohman and colleagues found that infants with BPD had significant differences in airway microbiome diversity compared to infants without BPD. Specifically, at the time of intubation, premature infants who developed BPD had lower bacterial diversity than those who did not develop BPD, as estimated by the number of observed species and the Shannon diversity index. This finding is supported by Tirone et al. [28] who also observed a slight reduction, although not statistically significant, in the Shannon diversity index in BALF samples from infants with BPD. Tirone and colleagues did not identify a characteristic microbiological signature linked to BPD development at either the genus or phylum level. In contrast, Lohman et al. [1] discovered distinct microbiome evolution patterns between infants with BPD and those without BPD, as revealed by their phylum-level analysis. The airway microbiome of infants with BPD exhibited an elevated presence of Firmicutes phyla alongside a reduction in Proteobacteria phyla, whereas infants without BPD showcased a comparatively diverse and stable microbiome. While *Acinetobacter* remained the predominant genus in both groups, its relative abundance decreased over time in the BPD group, with a concurrent rise in *Staphylococcus* and *Klebsiella* spp.

Lal et al. [20] also conducted a study on the airway microbiome using 16S rRNA sequencing. They analyzed tracheal aspirate specimens obtained from extremely low birth weight (EBLW) and full-term (FT) infants who underwent intubation and mechanical ventilation within the first six hours of life (due to respiratory distress syndrome, asphyxia and prenatal depression, or surgical indications). Tracheal aspirate samples were collected at the time of intubation and subsequently whenever tracheal suctioning was clinically indicated. The findings of this study indicated that the airway microbiota in infants with BPD showed decreased diversity and abundance (both alpha and beta diversity) and differed significantly from that of preterm infants soon after birth and full-term infants at a similar postmenstrual age. In contrast to Lohman et al.’s study [1], preterm infants with BDP exhibited an airway microbiome characterized by increased Proteobacteria and decreased Firmicutes and Fusobacteria compared to newborn full-term infants matched for PMA. Among Proteobacteria, the most abundant in infants with BPD were *Enterobacteriaceae*. *Lactobacillus* was found to be statistically significantly less abundant in the early airway microbiome of infants who later developed BPD, but the overall abundance of *Lactobacillus* was low. Furthermore, Lal et al. reported that the airway microbiome of both ELBW and FT infants at birth showed a predominance of Firmicutes and Proteobacteria on the first day of life, along with the presence of Actinobacteria, Tenericutes (including *Ureaplasma* spp.), Bacteroidetes, Cyanobacteria, Fusobacteria, and Verrucomicrobia. During the first 60 days of life, they noted a decrease in Firmicutes and increase in the abundance of *Proteobacteria* in the respiratory tract of ELBW.

In the study of Brewer and colleagues [3], the decrease in bacterial alpha diversity in the tracheal samples of intubated premature infants on DOL 3 to 7 was not associated with the diagnosis of BPD. Neither timing of OPC administration influenced the incidence of BPD.

Concerning the nasal microbiome in premature infants and its correlation with the development of BPD, we have identified a recent study conducted by Xu et al. [30] in 2022. This prospective observational study involved 28 premature infants (gestational age ≤ 30 weeks and birth weight 700–1550 g) divided into 13 infants with BPD and 15 controls. PCR amplification and 16S rDNA sequencing were performed on nasal swabs collected from the anterior nares at 1 and 3 weeks of postnatal age (5–7 DOL and 15–21 DOL, respectively). Xu and colleagues did not find a difference in nasal microbial diversity between preterm infants with BPD and the controls. They identified Firmicutes, Bacteroidetes, Proteobacteria, and Actinobacteria as dominant phyla and *Muribaculaceae*, *Escherichia*, *Staphylococcus*, and *Lachnospiraceae* as dominant genera in the nasal microbiome of all groups. The expression of *Prevotella* increased and *Caulobacter* decreased in the BPD group at both time points. *Prevotella* and *Caulobacter* were correlated with the severity of BPD. Additionally, *Prevotella* increased in infants on invasive mechanical ventilation, while *Caulobacter* decreased at both week 1 and week 3.

Recently, Selway et al. [26] studied the oral and respiratory microbiota of 50 very preterm neonates and discovered that the microbial diversity observed on buccal swabs during the first week of life (PT1) in preterm infants who developed BPD was lower compared to those infants who did not develop BPD, with a higher relative abundance of *Corynebacterium* (11.3% in BPD patients, 0.9% in not-BPD patients). Furthermore, they analyzed the diversity and composition of the microbiome in tracheal aspirate from premature infants requiring intubation. The most dominant amplicon sequence variants were identified as *Paenibacillus*, *Staphylococcus*, *Erythrobacteraceae bacterium K-2-3*, *Streptococcus anginosus*, *Ureaplasma*, *Burkholderiaceae*, and *Streptococcus agalactiae* sequences. *Ureaplasma* has been associated with an increased risk of developing BPD and was also identified as dominant taxa in buccal swabs (PT1) from infants with BPD who did not receive intubation.

## 4. Studies Excluded after Full-Text Screening

Quite a few articles were excluded after the full-text screening because the analysis of the airway microbiome was performed without using culture-independent molecular techniques and without reporting at least one alpha- or beta-diversity metric or a microbial profile [41]. Similarly, several studies were excluded because they exclusively analyzed the presence and quantity of specific organisms in samples from the airways, subsequently correlating them with certain outcomes, without examining the respiratory microbiota as a whole [42,43].

Three studies were excluded, although they were relevant to our scoping review, because the full text was not available or they were just congress abstracts. (1) The aim of the study by Gallacher and colleagues [44] was to investigate the microbial colonization of the lungs in intubated preterm infants at risk of developing BPD and to correlate these findings with pro-inflammatory cytokines through 16S rRNA sequencing on bronchoalveolar lavage samples. (2) Deshpande et al. [45] studied the gut and tracheal microbiome in preterm infants requiring mechanical ventilation who received probiotics (Lactobacillus acidophilus, Bifidobacterium bifidum). They showed that routine probiotic administration may alter the respiratory microbiota and consequently respiratory outcomes in preterm neonates. (3) Gorlanova et al. [46] compared the nasal microbiota of preterm and term infants at mean 45 weeks postmenstrual age and evaluated the impact of environmental factors on the nasal microbiota in preterm infants, reporting that nasal microbiota composition is associated with preterm birth.

We had to exclude from our literature review another interesting study [47] on the nasopharyngeal microbiota in preterm infants from birth until six months of age and the impact of antibiotics and hospitalization on it, as only the pre-print and non-peer reviewed version was available.

Several studies were excluded because, despite focusing on premature infants’ airway microbiota, the samples used to characterize the microbiome were collected at a non-neonatal age. Particularly, (1) Rofael et al. [48] characterized the airway microbiome in young adults born extremely premature, both with and without neonatal BPD, in comparison to matched term-born controls. They conducted gene sequencing on induced sputum from 92 individuals aged 19 years belonging to the EPICure study (51 extremely preterm and 41 controls) and reported a significant dysbiosis in the airway microbiome associated with extremely preterm birth, which correlated with a forced expiratory volume in the first second (FEV1). (2) Perez et al. [49] conducted a study on the nasopharyngeal microbiome of preterm and full-term infants aged between 6 months and 2 years. The taxonomic profile obtained in samples from preterm infants differed slightly from one of the full-term controls. They found *Moraxella* and *Burkholderia* being predominant, mixed with *Streptococcus*, *Neisseria*, *Staphylococcus*, *Janthinobacterium*, and *Haemophilus*, and the within-group heterogeneity was very high. In full-term infants instead, *Streptococcus* was predominant, with low dissimilarities within the group. At the phylum level, preterm infants showed an increase in Proteobacteria (a large group of Gram-negative bacteria) and a decrease in Firmicutes (Gram-positive bacteria, such as *Clostridium* and *Lactobacillus*). To assess the functionality of the respiratory microbiota in modulating airway inflammation and respiratory symptoms during viral infections in preterm infants, samples were obtained during acute infection and at a follow-up visit. Upon comparing the paired samples, Perez and colleagues found that the preterm airway microbiota was less resilient during a rhinovirus (RV) infection, exhibiting a high within-group dissimilarity in preterm infants at baseline, which becomes higher in RV samples. This suggests an “unstable” microbiome prone to widespread changes in bacterial diversity during minor viral illnesses.

## 5. Discussion

The exploration of the airway microbiome in preterm infants has recently emerged with novel insights into its distinctive characteristics, contrasting dynamics with term infants; its correlation with adverse outcomes, such as BPD; and thus potential therapeutic interventions.

About the distinctive features of preterm infants’ airway microbiome, our analysis highlights notable differences between the airway microbiota of preterm and term infants. Studies consistently report that gestational age plays a pivotal role in characterizing the early microbial colonization [18,19,21,23]. In fact, gestational age at birth, rather than postnatal age, appears to influence the characteristics of the airway microbiota at birth. Preterm birth at lower gestational ages seems to result in reduced diversity in the microbiota, which is dominated by *Staphylococcus*. Later, a convergence of microbiota profiles around 50 weeks PMA in both preterm and term-born infants was noted, suggesting a developmental trajectory towards a shared pattern [19]. However, significant disparities persist during the early stages, emphasizing the influence of prematurity on the initial microbial landscape.

Many studies found an association between the nearest and farthest anatomical sites of a sampling microbiome and found strong interconnections [3,19,31,32,33]. Similarly, a recent scoping review [50] provides valuable insights into the oral microbiome of preterm infants, complementing our focus on the airway. The review emphasizes the interconnectedness of oral and airway microbiomes, hinting at potential seeding reservoirs for respiratory microbes. This interplay adds a layer of complexity to our understanding and underscores the need for a comprehensive approach in studying the microbial landscape of preterm infants.

Our scoping review, in addition to broadening the scope of analysis, updates the systematic review by Pammi and colleagues [2], which examined the available evidence up to 2017 regarding the airway microbiome and BPD in preterm neonates. In our study, a significant portion of the reviewed literature delves into the association between the airway microbiome and the development of BPD in preterm infants. The findings suggest the possibility of a complex interplay between microbial diversity, composition, and the risk of BPD, even if this should still be clarified. In some of the included studies, infants diagnosed with BPD exhibit distinct microbial evolution patterns, marked by lower diversity at the time of intubation [1,20,28,29] and variations in phylum-level abundance [1,24,26,34], but the impact of these variations on clinical management needs further clarification. The prevalence of specific genera, such as *Ureaplasma*, stands out as potential indicators of BPD risk; the causal relationship between organism and disease also may need further investigation. The observed microbial signatures offer valuable insights into the intricate relationship between early microbial colonization and the subsequent respiratory health of preterm infants. The potential impact of dysbiosis in determining an outcome such as BPD is supported by other longer-term or broader-spectrum evidence, such as the evidence of dysbiosis correlated with reduced FEV1 in preterm adults [48] or associated with cystic fibrosis, lower respiratory tract infections, and asthma [51,52].

Understanding the impact of perinatal factors on the airway microbiome is crucial for contextualizing our findings. The delivery mode emerges as a key determinant, with vaginally born preterm neonates displaying different microbial profiles compared to those delivered via cesarean section. The influence of antibiotics and antifungals on microbiota composition, although explored to a varying extent across studies, adds another layer of complexity. Interestingly, our review did not reveal consistent patterns regarding the effect of antibiotic exposure on microbial diversity, suggesting multifaceted interactions between antibiotics, microbial communities, and respiratory outcomes.

Identifying potential therapeutic interventions to modulate the airway dysbiosis in preterm infants emerges as a critical avenue for further research. Current evidence suggests a correlation between early T cell phenotypic differentiation, microbiota exposures, and the risk of respiratory compromise. This age-independent microbiota–T cell axis highlights the potential impact of prenatal exposures, such as antibiotics or infections, on disrupting developmental trajectories. Exploring interventions to manipulate the airway microbiome, such as probiotics or targeted antibiotics, may potentially provide avenues to reduce the risk of adverse respiratory outcomes. However, as per the existing literature, there are no studies demonstrating this effect.

In conclusion, our review delves into the intricate subject of the respiratory microbiome in preterm infants and its association with long-term respiratory outcomes. It is clear that there is ample room for further exploration and investigation in this field, and presently, there are no studies in the literature that provide comprehensive clarification on the matter.

## 6. Limitations, Knowledge Gaps, and Directions for Future Research

There were some limitations to the studies that we reviewed.

In the examination of endotracheal aspiration across multiple studies, with the exception of one [32], the airway microbiota was consistently sampled through endotracheal tubes. Consequently, the ability to draw conclusions concerning the evolution of the airway microbiome in free-breathing infants is limited (such as in preterm infants on continuous positive airway pressure or with a high-flow nasal cannula). An additional concern lies in the potential for endotracheal tube aspiration to overemphasize bacteria that colonize the endotracheal tube itself rather than accurately representing the microbial composition of the broader airway or lungs. This introduces a potential bias in the interpretation of the airway microbiome dynamics in the studied population.

Another potential limitation concerns studies that analyze the microbiome in correlation with the development and severity of BPD. On one hand, the definition of BPD is still controversial, and therefore, the use of different definitions may compromise the comparison of results across various studies. On the other hand, although significant, the clinical studies mentioned earlier are inherently observational; thus, they do not furnish any evidence supporting a causal relationship between alterations in the airway and lower respiratory microbiomes and the clinical course of affected infants. For this reason, recently, there has been an emerging interest in preclinical studies analyzing the role of the microbiome in lung development [53]. One of the latest studies, for example, confirmed the role of airway dysbiosis on the risk of developing a more severe BPD phenotype by influencing the expression of local antioxidant responses, which could be ameliorated by supporting colonization with *Lactobacillus* [54]. Thus, more studies are mandatory on this argument to better understand targets of future microbiome-based therapeutics for respiratory disease.

In the analysis of the neonatal respiratory microbiome, advancements such as the utilization of 16S rRNA sequencing, a method employed in the majority of included studies, have significantly enriched our understanding. However, it is crucial to acknowledge a limitation in these methodologies, as they do not distinguish between viable and dead bacteria [3]. On the other hand, a metagenomic approach employing shotgun sequencing offers increased granularity in a sample composition analysis. This approach enables the identification of bacterial species or strains and provides information on antimicrobial resistance mutations. Further, RNA metagenomic profiling can offer additional insights into microbial activity and the metabolism of microbiomes [55,56]. Future studies on the newborn airway microbiome that integrate meta-taxonomic methods (such as 16S rRNA targeted sequencing) and metagenomic approaches would yield additional evidence, potentially enhancing treatment approaches for adverse respiratory outcomes, such as BPD and asthma [51].

The evolving landscape of research in preterm infants’ airway microbiome presents exciting opportunities for innovative investigations and potential therapeutic interventions. Characterizing the lung microbiome in premature infants opens avenues for novel treatments aimed at preventing dysbiosis. Strategies such as oral or inhaled probiotics, prebiotics, dietary manipulation, the judicious use of antibiotics, therapies targeting proinflammatory bacteria, and the development of bacterially derived drugs for inflammation modulation and bacterial overgrowth are all promising areas for exploration. Recognizing that bacterial colonization likely initiates in utero, addressing maternal dysbiosis through probiotics and prebiotics during pregnancy emerges as a prospective strategy. Moreover, delving deeper into the gut–lung axis and exploring the role of extra-pulmonary microbes in respiratory disease development stands as an imperative avenue for further investigation [57,58].

This holistic approach to understanding and influencing the airway microbiome in preterm infants holds great potential for advancing both preventive and therapeutic interventions in neonatal respiratory health. In recent times, advancements in multi-omics technologies, encompassing genomics, transcriptomics, proteomics, and metabolomics, have expanded the potential for discovering and potentially developing more informative biomarkers for application in clinical practice [59]. These technologies could aid in the understanding of mechanisms as well as preventative strategies in this neonatological field.

## 7. Conclusions

The exploration of the airway microbiome in preterm infants represents a dynamic and evolving field with far-reaching implications for neonatal health. The synthesis of the current literature highlights the intricate interplay between microbial colonization, respiratory outcomes, and potential therapeutic interventions. As we navigate through the complexities of the preterm airway microbiome, it becomes evident that future research should delve deeper into the mechanistic underpinnings of dysbiosis and its impact on respiratory health. Addressing methodological challenges, incorporating advanced technologies, and fostering collaborative, multi-center studies will be essential in advancing our understanding. The promising avenues of intervention, from probiotics to targeted antibiotic therapies, underscore the potential to shape and modulate the airway microbiome for improved neonatal outcomes. With these insights, the trajectory of research in preterm infants’ airway microbiome not only contributes to the refinement of clinical practices but also lays the foundation for personalized and effective strategies in promoting respiratory health in this vulnerable population.

## Figures and Tables

**Figure 1 nutrients-16-00465-f001:**
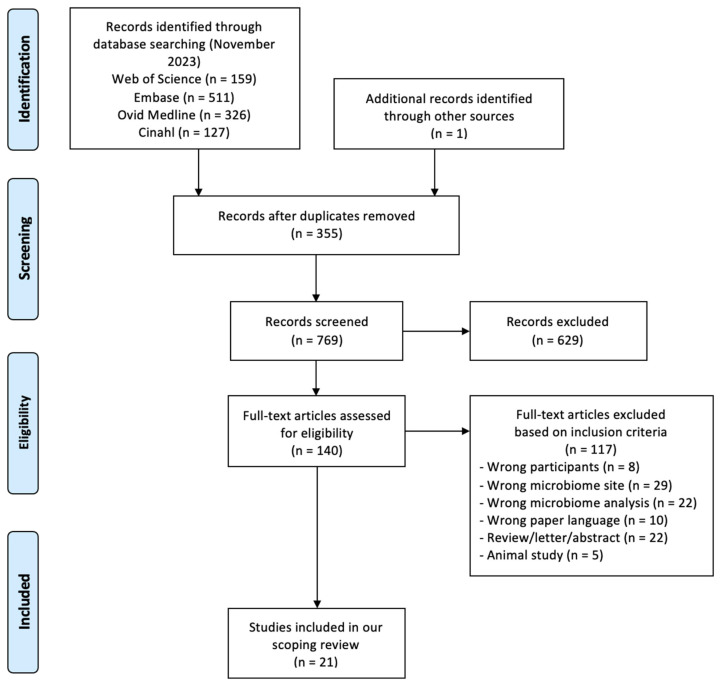
PRISMA 2009 research method flow diagram.

**Table 1 nutrients-16-00465-t001:** Characteristics of included studies.

Study	Country	Population	N° of Neonates	Microbiome Site	Aim
Brewer 2021 [3]	USA	GA ≤ 32 weeks, birth weight (BW) < 2000 g and intubated for respiratory distress syndrome	42	Tracheal aspirates (TA) on days 3–7, oral aspirate DOL 3	Oral vs tracheal microbiota during the first week of life, effects of OPC administration on microbial diversity or leukocyte inflammatory activity in the lung
dos Anjos Borges 2023 [17]	Germany	GA ≤ 34 weeks born from mothers with preterm premature rupture of membranes (PPROM)	89	vaginal swab—umbilical cord blood (UC), rectal swabs (RE), and pharyngeal swabs (PH) at delivery-meconium sample prior to 48 h of life	Correlation between vaginal PPROM mothers’ microbiome and pharyngeal/umbilical cord/meconium/rectal neonatal microbiota
Gallacher 2020 [18]	UK	GA ≤ 32 weeks and intubated	55	TAF, BAL, and NPA daily during the first week then twice weekly or until extubation, stool weekly until 28 DOL	Temporal modification of bacterial communities, gut–lung axis. Association between the bacterial communities and lung inflammation.
Grier 2018 [19]	USA	Preterm and full-term infants	82	Rectal, nasal, throat swabs weekly during admission, monthly after discharge until 1 year of corrected age (CA)	Temporal modifications of the early microbiota and associations of microbiota across body sites. Differences between term and preterm infants’ microbiota.
Lal 2016 [20]	USA	ELBW preterm infants, full-term infants	51	TA at intubation and at clinically indicated tracheal suctioning	Differences in preterm and FT infants’ airway microbiota and association with risk of developing BPD
Lohman 2014 [1]	Texas, USA	GA ≤ 32 weeks and intubated in the first 24 h	25	TA at intubation and 3, 7, and 28 DOL	BPD
McDavid 2022 [21]	USA	GA 23–42 weeks	267	Nasal swab or rectal swab weekly or monthly in the first year of life	Correlation between immune system, microbiota and respiratory morbidity
Mourani 2011 [22]	Colorado, USA	GA ≤ 34 weeks, BW 500–1250 g, mechanical ventilation (MV) for at least 21 days	10	TA within 72 h of birth and 7, 14, and 21 DOL	BPD
Pattaroni 2018 [23]	Switzerland and New Zealand	GA 23–41 weeks with endotracheal intubation for elective surgery or respiratory support	52	TA at postnatal age 1 day–1 year	To identify the early-life bacterial colonization pattern of the lower airways in relation to the maturating immune system during the first year of life.
Payne 2010 [24]	UK	BW < 1300 g requiring MV for at least 24 h	55	Gastric fluid aspirate, Endo-TA (ETA) within 24 h of life	BPD
Rosenboom 2023 [25]	Germany	GA 24–33 weeks, full-term infants	24	Oropharyngeal swabs at week 1 of life, 1–9–15 months of life	Longitudinal development of the airway metagenome of preterm infants during the first 2 YOL.
Selway 2023 [26]	Australia	Preterm infants, full-term infants, adult	64 and 16 adults	Buccal swab 2–12 DOL and 36 weeks GA, TA at intubation	Compare preterm–term infant–adult microbiome and BPD/sepsis
Stressman 2010 [27]	UK	GA ≤ 31 weeks, intubated at birth and received surfactant therapy	8	ETA in the first week of life	to investigate the use of culture-independent molecular profiling methodologies to identify potential etiological agents in neonatal airway secretions
Tirone 2022 [28]	Italy	GA ≤ 30 weeks intubated in the first 24 h of life and their mothers	29 and 26 mothers	Mothers’ vaginal swab, newborns’ BAL fluid at 1–3–7 DOL and meconium specimen	Relationship between maternal vaginal microbiota and neonatal lung and intestinal microbiota; association between neonatal lung/meconium microbiota and BPD
Xu 2022a [29]	China	GA < 34 weeks and intubated in the first 24 h	28	TA day 1 and 7	Mild, moderate, severe BPD and no BPD
Xu 2022b [30]	China	GA ≤ 30 weeks	28	Nasal swab 5–7 DOL and 15–21 DOL	BPD
Yang 2021 [31]	China	GA 26–32 weeks	13	Stool samples and pharyngeal swabs 1–28 DOL	Gut–lung axis
Yao 2023 [32]	China	GA 32–35 weeks	40	Endotracheal suction at birth and stool specimen from the first fetal stool	Diversity and community structure in the bacterial microbiome of the airways and intestines of preterm twin and singleton neonates
Young 2020 [33]	UK	GA < 26 weeks	7	Oral and endotracheal secretions, stool, and breast milk over the first 60 days of life	Characterize the development of microbiota
Wagner 2017 [34]	USA	GA ≤ 34 weeks and BW 500–1250 g and MV	152	TA at enrollment, 7, 14, and 21 DOL (±48 h)	BPD
Wagner 2019 [35]	USA	GA ≤ 34 weeks and BW 500–1250 g and MV	71	TA at 7 ± 48 h DOL	Relationship between prenatal events, postnatal airway host response and microbiota, and clinical outcomes

## Data Availability

Data sharing not applicable.

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
