# Peer review of "Preterm Infants’ Airway Microbiome: A Scoping Review of the Current Evidence"

_nutrients, 2024, doi:10.3390/nu16040465_

Round 1

Reviewer 1 Report

Comments and Suggestions for Authors

Author Response

  1. Mention is made that all fetuses experience initial colonization during the prenatal

period. This remains somewhat controversial, and many would beg to differ with

that statement. There is a significant literature on the Sterile Womb Hypothesis

and it is suggested that the authors provide this more consideration.

We included a reference that underlines that the 'sterile womb hypothesis' has not been fully overcome, despite recent literature suggesting the 'in utero colonization hypothesis’. [Perez Munos 2016]

  1. Lines 54-65-- Introduction--- it is not clearly written how this interaction between

microbiota, lung epithelial cells and leukocytes trigger the activation of toll like

receptors and how this contributes to the development of BPD. The references

provided do not appear to provide a clear mechanism for this.

We thank the Reviewer for the comment. We added in this paragraph the citation of the paper written by Følsgaard NV, and colleagues, which clearly explain how colonization of the neonatal airways with pathogenic bacteria could lead to atypical inflammation, chronic inflammation and thus to BPD.

  1. Lines 128-130—This is confusing. Please clarify and expand to provide a more

detailed explanation of what occurs during this convergence.

We tried to better explain this section.

  1. There is considerable information provided about taxonomic differences over

time and in certain studies, but little is mentioned about significance of these

microorganisms. This becomes highly important when discussing airway

responses as in lines 225-250

We added some information and references.

  1. PICRUSt analysis---lines 313-315---nothing is mentioned about the limitations of

this analysis and more sophisticated techniques that could be used.

It was not our intention to claim that PICRUSt is the best methodology, but we aimed to provide a description of what was reported in the study. PICRUSt certainly has several limitations, for example that the PICRUSt’s ability to detect patterns depends on the input data used and that only 16S marker gene sequences corresponding to bacterial and archaeal genomes are currently included.

We added a reference [Langille et al 2013] in which is well explained the limitations of PICRUSt (and that is present in the bibliography of Grier et al 2018).

  1. Discussion---are there actual strong correlations between BPD development and

microbiome. Furthermore, even if there are strong correlations between

taxonomy and disease, is there evidence of causality? Which came first, the

disease or the organism?

We modified the paragraph as follows: “In our study, a significant portion of the reviewed literature delves into the association between the airway microbiome and the development of BPD in preterm infants. Findings suggest the possibility of a complex interplay between microbial diversity, composition, and the risk of BPD, even if this should be still clarified. In some of the included studies, infants diagnosed with BPD exhibit distinct microbial evolution patterns, marked by lower diversity at the time of intubation [1,25,27,31] and variations in phylum-level abundance [1,26,30,33], but the impact of these variations on clinical management needs further clarification. The prevalence of specific genera, such as Ureaplasma, stands out as potential indicators of BPD risk, also if the causal relationship between organism and disease needs further investigation. “

  1. The current evidence does not provide specific evidence as to which

interventions might be of benefit.

We agree with your comment. Discussing which possible interventions might be of benefit was not the aim of our review. In ‘Discussion’: “Exploring interventions to manipulate the airway microbiome, such as probiotics or targeted antibiotics, may potentially provide avenues to reduce the risk of adverse respiratory outcomes. However, as per the existing literature, there are no studies demonstrating this effect.”

  1. No mention is made about integrated multiomics and systems-based approaches

for biomarker discovery. This could aid in the understanding of mechanisms as

well as preventative strategies.

We added a sentence about this in “Limitations, Knowledge Gaps, and Directions for Future Research “.

Reviewer 2 Report

Comments and Suggestions for Authors

The aim of this extensive review was to investigate and synthesize current evidence on airway microbiome of preterm infants and to correlate it with the development of BPD. 

The review is well performed and well written. Messages are clear. 

Nevertheless I think the manuscript should undergo some minor revisions before publication. First of all I htink the authors should mitigate their message: in fact current evidence does not demonstrate a clear correlation between the different microbiomes and outcomes. Although it would be expected to have some differences between between neonates with different microbiomes, no clear evidence exists to date that the microbiome has an influence on the outcome of the babies. This review in my opinion highlights the lack of clear relation between microbiome and outcome. This could be an opportunity to rethink current research: should we design differently our research on the relation microbiome/diesase?  

Here are my point by point criticisms: 

Line 450: Could the authors add the number of neonates involved in the study of Selway et al.

It would be useful to add the number of neonates involved in each study in the Table 1. In fact some studies have a really small number of infants (8) versus other studies have a much bigger cohort. 

Line 567: "Exploring interventions to  manipulate the airway microbiome, such as probiotics or targeted antibiotics, could offer avenues for mitigating the risk of adverse respiratory outcomes." This is an assumption the authors are making but without a real scientific basis, as till now the studies performed do not demonstrate a difference in the incidence of BPD when babies are treated for ureaplasma, nor treated with antibiotic prophylaxis.

Line 481: "They showed that routine probiotic administration may alter the respiratory microbiota, 481 and consequently respiratory outcomes in preterm neonates." Could the authors add the type of probiotic that was administered? 

Line 542: "Findings collectively  suggest a complex interplay between microbial diversity, composition, and the risk of BPD. Notably, infants diagnosed with BPD exhibit distinct microbial evolution patterns, marked by lower diversity at the time of intubation and variations in phylum- level abundance ." 

The authors should be cautious in affirming this concept, as in fact the different studies did not observe a characteristic microbial fingerprint of neonates that will later develop BPD. The fact that there is a reduced diversity in infants developping BPD will really have an impact on our management of these infants? In the studies that observed dysbiosis in preterm adults it is not clear if this dysbiosis in the cause or the effect of prematurity.

Line 572: Conclusion: again I would be careful in stating that microbiome influences respiratory outcomes, as the studies performed till now did not reveal a specific pattern of microbiome associated to BPD. I think this well performed review does not "shed the light" but on the contrary demonstrates that research is still to be done in this complex field of microbiome interaction with complex diseases. 

Author Response

  • Line 450: Could the authors add the number of neonates involved in the study of Selway et al.

We added the number of involved neonates (Line 450-451)

  • It would be useful to add the number of neonates involved in each study in the Table 1. In fact some studies have a really small number of infants (8) versus other studies have a much bigger cohort. 

We added the number of neonates involved in each study in Table 1

  • Line 567: "Exploring interventions to  manipulate the airway microbiome, such as probiotics or targeted antibiotics, could offer avenues for mitigating the risk of adverse respiratory outcomes." This is an assumption the authors are making but without a real scientific basis, as till now the studies performed do not demonstrate a difference in the incidence of BPD when babies are treated for ureaplasma, nor treated with antibiotic prophylaxis.

We modified: “Exploring interventions to manipulate the airway microbiome, such as probiotics or targeted antibiotics, may potentially provide avenues to reduce the risk of adverse respiratory outcomes. However, as per the existing literature, there are no studies demonstrating this effect.”

  • Line 481: "They showed that routine probiotic administration may alter the respiratory microbiota, 481 and consequently respiratory outcomes in preterm neonates." Could the authors add the type of probiotic that was administered? 

We added it

  • Line 542: "Findings collectively  suggest a complex interplay between microbial diversity, composition, and the risk of BPD. Notably, infants diagnosed with BPD exhibit distinct microbial evolution patterns, marked by lower diversity at the time of intubation and variations in phylum- level abundance ." 

The authors should be cautious in affirming this concept, as in fact the different studies did not observe a characteristic microbial fingerprint of neonates that will later develop BPD. The fact that there is a reduced diversity in infants developping BPD will really have an impact on our management of these infants? In the studies that observed dysbiosis in preterm adults it is not clear if this dysbiosis in the cause or the effect of prematurity.

We changed the sentences as follows “Findings suggest the possibility of a complex interplay between microbial diversity, composition, and the risk of BPD, even if this should be still clarified. In some of the included studies, infants diagnosed with BPD exhibit distinct microbial evolution patterns, marked by lower diversity at the time of intubation [1,25,27,31] and variations in phylum-level abundance [1,26,30,33], but the impact of these variations on clinical management needs further clarification “

  • Line 572: Conclusion: again I would be careful in stating that microbiome influences respiratory outcomes, as the studies performed till now did not reveal a specific pattern of microbiome associated to BPD. I think this well performed review does not "shed the light" but on the contrary demonstrates that research is still to be done in this complex field of microbiome interaction with complex diseases. 

We modified it as follows: “In conclusion, our review delves into the intricate subject of the respiratory microbiome in preterm infants and its association with long-term respiratory outcomes. It is clear that there is ample room for further exploration and investigation in this field, and presently, there are no studies in the literature that provide comprehensive clarification on the matter.”

Round 2

Reviewer 1 Report

Comments and Suggestions for Authors

No further concerns